# Impact of food insecurity with hunger on mental distress among community-dwelling older adults

**Razak M. Gyasi**[1]*, **Bernard Obeng**[2], **Joseph Y. Yeboah**[3]

**1** African Population and Health Research Center (APHRC), Nairobi, Kenya, **2** Department of Sociology and Social Work, Kwame Nkrumah University of Science and Technology, Kumasi, Ghana, **3** Department of Geography and Rural Development, Kwame Nkrumah University of Science and Technology, Kumasi, Ghana

* RGyasi@aphrc.org, RGyasi@LN.hk

**Data Availability Statement:** Upon another contact with the Ethics Committee of the Lingnan University, the data for this paper has currently been released to me and made freely available for public access and use without any institutional

## Abstract

### Background

Hunger frequently and persistently occur in older populations in low-income countries especially in sub-Sahara Africa. The aim of this study was to examine the associations between food insecurity with hunger and psychological distress among older people in Ghana.

### Methods

A total of 1200 individuals aged ≥50 years were recruited during 2016/2017 Ageing, Health, Psychological Well-being and Health-seeking Behavior Study. Associations between psychological distress (assessed with the Kessler Psychological Distress Scale) and hunger (assessed with a 30-day subjective scale) were evaluated using linear regression modeling.

### Results

The overall prevalence of food insecurity was 36% with approximately 27% and 9% respectively for moderate and severe levels of hunger whilst the mean score of psychological distress was 9.5 (±4.10). Persons experiencing moderate hunger (β = 0.71, SE = 0.160, $p <$ 0.001) and severe hunger (β = 1.81, SE = 0.280, $p <$ 0.001) significantly reported increased psychological distress outcome compared to those without hunger. These associations varied between women (β = 1.59, SE = 0.359 $p <$ 0.001) and men (β = 2.33, SE = 0.474, $p <$ 0.001) as well as 50–64 age group (β = 1.48, SE = 0.368, $p <$ 0.005) and 65+ age group (β = 2.51, SE = 0.467, $p <$ 0.001).

### Conclusions

The results suggest that experiencing hunger is associated with psychological distress and the effect may be aggravated with advancing age and in men. These findings may inform social policy initiatives and health programmatic interventions for older people exposed to food insecurity.

restrictions. In this regard, I have the authority to display the data freely without contacting the copyright holders for registration and access. I would be able to submit the data to the journal's website upon acceptance of the paper

**Funding:** This work was supported by Lingnan University, Hong Kong [RPG 1129310] to RMG (https://www.ln.edu.hk/about-lu/introducing-lingnan). The funders had no role in study design, data collection and analysis, decision to publish, or preparation of the manuscript.

**Competing interests:** The authors have declared that no competing interests exist

## Introduction

Social and economic dynamics have been recognized as important factors in coping with stressful life course circumstances such as hunger and food insecurity [1–3]. However, this role has rarely been examined in relation to psychological distress in later life, even though older people are more at risk of moderate-to-severe food insecurity and mental distress [4]. Given that low- and middle-income countries (LMICs) will be home to more than four in five older people by 2050 with concomitant mental-related infirmities [5], the risk of psychological distress may be greater following exposure to food insecurity in older age. It is, thus, imperative to understand the role of food insecurity with hunger in psychological health in later life.

Hunger has been theorized as a disquiet associated with an involuntary absence of food [6]. *The Global State of Food Security and Nutrition* report [1] indicates that more than 820 million people globally are hungry and the vast majority are older people residing in sub-Saharan Africa. This is, largely, due to multiple socioeconomic impacts such as financial vulnerabilities, lack of retirement protection and gradual breakdown of filial piety [5, 7] highlighting the mammoth challenge of triumphing the *Zero Hunger* target by 2030. *The National Foundation to End Senior Hunger* report shows that approximately one in six older Americans face the threat of hunger [8]. However, data on hunger in older age is almost non-existent in sub-Saharan Africa and Ghana in particular.

Studies have shown that the experience of being hungry or food insecure remains a major source of a wide-ranging lethal health outcomes including physical and mental health problems among the general population [9–11]. Focusing on mental problems, anxiety [12], depression [13], suicidal ideation [14–16] and cognitive impairment [17, 18] have be related to hunger/food insecurity. While the linkage between hunger and psychological distress is an emerging public health concern, there is a paucity of contemporary studies assessing this relationship among older populations in LMIC settings.

Previous studies show that men and women and specific age cohorts may differ in health and the mechanisms that predict their psychological state [5, 19]. The *differential vulnerability thesis* [20], indicates that gender and age differences exist in the vulnerability to social determinants of health such as hunger partly due to variant socioeconomic deprivations among genders and with advancing age. For example, compared to men and older age groups, women and younger age group respectively tend to invest in stronger constellation of social networks [19, 21] which can protect against stress and food insecurity and subsequently enhance their psychological wellbeing [22]. Yet, it is unclear how gender and age differences impact the association of hunger with psychological distress. The aim of this study was to investigate: 1) the effect of hunger on psychological distress in older age in Ghana, and 2) whether the associations of hunger with psychological distress are differentiated by gender and age. We hypothesize that persons exposed to hunger will display worse psychological distress (hypothesis 1). We also expect that the association between hunger and psychological distress will be marked in men and with advancing age (hypothesis 2).

## Methods

### The survey and sample

Data for this study came from an Aging, Health, Psychological Wellbeing and Health-seeking Behavior Study conducted in Ghana between July 2016 and February 2017. The larger, original study aimed at generating comprehensive data on socioeconomic, health and wellbeing profile of older people in Ghana and provided a basis for comprehensive analysis of many facets of aging in the LMICs. This was a probability-based sample consisting of adults aged ≥50 years. This eligibility

criterion was developed given that chronological time has little or no relevance in conceptualizing old age in many parts of SSA [5]. Several recent gerontology studies have adopted the above categorization to define older people, including the Minimum Data Set project on aging and many other regional studies, such as the WHO's SAGE study in LMICs, including Ghana [23].

A multi-stage stratified cluster sampling procedure was followed [24]. Details of selection procedure have been reported elsewhere [21, 22, 25–28]. Three sub-regional zones (based on their geographic uniqueness) were formed as strata and used as primary sampling units. Two districts were randomly selected from each sub-regional zone and delimited into rural and urban areas based on the Ghana Statistical Service's [29] classification. The sample size was estimated, assuming 5% margin of error, 95% confidence interval, 1.5 design effect, 5% type 1 error, 15% type 2 error, and a default prevalence of 0.5. The required sample size was 901, but considering a 35% non-response, the final proposed sample size for this study was approximately 1219. The statistical power calculation revealed that the sample size had 85% power to detect an odds ratio of $\geq 2$. In total, 1247 older persons were selected through systematic random sampling procedure. Of the 1247 approached, 1219 (97.8%) were eligible to participate but 19 of them declined yielding an overall participation rate of 98.4% ($N$ = 1200).

The survey questionnaire was initially developed in English, translated into Asante Twi (the principal dialect in the study area) and back translated into English with reconciliation of discrepancies for quality control of the translation procedure following WHO translation guidelines for assessment instruments [30]. Face-to-face interviews were conducted using interviewer-administered questionnaire.

## Ethic statement

Study protocol was approved by the Committee on Human Research Publication and Ethics, School of Medical Sciences, Kwame Nkrumah University of Science and Technology and Komfo Anokye Teaching Hospital, Kumasi, Ghana (Ref: CHRPE/AP/507/16). Ethics approval was also granted by the Research Ethics Committee of Lingnan University, Hong Kong, before interviews began. Study participants provided written informed consent, which was either signed or thumbprinted based on the choice of the participant, mainly based on their literacy levels, after briefing them on the research aims, procedures and the voluntary nature of their participation.

## Assessment of variables

*Psychological distress* was the outcome variable The 10-item Kessler Psychological Distress Scale (K10) was used to assess non-specific psychological distress, composed of items measuring psychological and physiologic symptoms of anxiety and depression in the previous four weeks [31]. This scale was developed as a screening tool for psychological distress in the general population. Each item has five response scale: 1) none of the time, 2) a little of the time, 3) some of the time, 4) most of the time, and 5) all of the time. Total score of the K10 ranges from 10 (no distress) to 50 (severe distress). Studies have shown that the K10 is psychometrically valid and appropriate for use in indigenous and general populations in SSA and Ghana [32].

The key exposure variable of interest, *hunger* was assessed with the item "During the past 30 days, how often did you go hungry because there was not enough food in your home?" The item was scored on a five-point response scale 1 = never, 2 = rarely, 3 = sometimes, 4 = most of the time, 5 = always. We transformed the response options into 0 = never/rarely (no food insecurity), 1 = sometimes (moderate food insecurity), 2 = most of the time/always (severe food insecurity) [33]. Whilst moderate food insecurity often suggests a compromised quality and quantity of food consumed, severe food insecurity has been described as reduced food intake and disrupted eating patterns particularly in later life [16].

The analysis adjusted for a number of potential confounders which have been reported to link both psychological distress and food insecurity in low-income settings [25, 34]. These included age (in years), gender, household size, level of education, employment status, income levels (in Cedis), rural/urban residence, marital status, social support, self-rated health, functional status, chronic conditions, alcohol intake, and physical activity.

Age was a continuous variable and also categorized into middle age (50–64 years) and older age (≥65 years). Gender was treated as a dummy variable with 1 if female and 0 if otherwise. Spatial residence was dichotomous with 1 if urban and 0 if otherwise. Marital status, recoded from four original responses (married/ cohabiting, divorced/separated, widowed, never married) was put into two categories: 1 if unmarried and 0 if married. Education reflected three schooling levels: 0 = never/basic, 1 = secondary, 2 = higher. Employment status was dichotomized into 1 if employed, and 0 if otherwise, while household income and household size were scored as continuous variables. We assessed social support with frequencies of contact with family or close friends and social participation.

Self-rated health was assessed with an item asking respondents to self-rate their health with a five-point response scale: 1 = excellent, 2 = very good, 3 = good, 4 = fair and 5 = poor which were later collapsed into four-options. Chronic conditions were assessed with a question, "Did a doctor or health professional ever told you that you had...?" The list of conditions included hypertension, diabetes, respiratory diseases, cancers, stroke, chronic kidney diseases, asthma, arthritis, depression and insomnia. Functional status was assessed on level of performance of five-item of activities of daily living (ADL) that are required to take care of oneself such as bathing, toileting, eating, bathing and dressing and getting in and out of bed. These items were recorded on a four-point scale: 1 = not limited at all, 2 = less limited, 3 = somewhat limited, and 4 = much limited [35]. Alcohol use was assessed with a no/yes item if they consumed any drink that contained alcohol such as beer, hard wine, spirit, over the past 30 days.

## Statistical analysis

The statistical analyses were done using SPSS v.21.0 (IBM, Armonk, NY). While univariate statistics were executed to contextualize the sample, we conducted bivariate associations between hunger and psychological distress outcome by Chi-squared test (results not shown) given the categorical state of variables. Multiple linear regressions were conducted to assess the associations between hunger and psychological distress in the pooled sample and also by gender- and age-specific groups. This was crucial because the risk factors of psychological distress have been noted to differ by gender and between mid-life and later-life [26].

Model 1 evaluated the crude association of hunger with psychological distress and Model 2 introduced an adjustment for potential confounders in the regression model. Because food insecurity levels may differ between rural and urban areas [34] we performed interaction analysis between geographical variation and food insecurity (spatial location × hunger) in further sensitivity analysis. We checked for multicollinearity by computing the Variance Inflation Factor (VIF). In our analysis, none of the VIF scores exceeded the value of 2.5, indicating no multicollinearity. Beta values with robust standard errors were reported with $p$-value less than 0.05 as statistically significant.

## Results

### Sample characteristics

Table 1 presents the descriptive statistics of the sample. Mean age was 66 years (SD = ±12) and the majority were women (63%, 95%CI: 95%CI: 60.5–66.0%). Most of the respondents were not married (57%), lived in urban areas (55%), were unemployed (86%) and had lower levels

of education (86%). Incomes were generally low with an average of GH¢308 (US$59). Approximate average social support (6 [SD = ±2.7]), physical activity (9 [SD = ±4.4]) and functional impairment (14 [SD = ±5.1]) were reported. More importantly, moderate hunger (27%), and severe hunger (9%) were revealed. Psychological distress was significantly higher among those experiencing severe hunger compared to those without hunger (15.1% vs 3.7%, $p < 0.001$).

## Main regression models

Linear regressions (Table 2) revealed that hunger (food insecurity) was an independent predictor of psychological distress even after accounting for potential socioeconomic and health-related confounders. The initial crude analysis (Model 1) showed that compared to no hunger, moderate levels of hunger (β = 0.95, SE = 0.136, $p < 0.001$) and severe levels of hunger (β = 1.83, SE = 0.136, $p < 0.001$) were significantly associated with increased risk of psychological distress outcome. These associations persisted after full adjustment. In Model 2, moderate levels of hunger (β = 0.71, SE = 0.160, $p < 0.001$) and severe levels of hunger (β = 1.81, SE = 0.280, $p < 0.001$) significantly associated with increased psychological distress score compare to no hunger. In addition, age, not married, urban dwelling, lower social support and worsening SRH outcome significantly predicted psychological distress. In a confirmatory analysis, we

**Table 1. Descriptive statistics of the study sample.**

| Variable | | | Valid N | (%) |
|---|---|---|---|---|
| Psychological distress score (mean score) | | | 9.54 | (±4.10) |
| *Hunger* | | | | |
| | Never/rarely (No food insecurity) | | 773 | (64.4) |
| | Sometimes (Moderate food insecurity) | | 321 | (26.8) |
| | Most of the time/always (Severe food insecurity) | | 106 | (8.8) |
| Age (in years) (mean score) | | | 66.15 | (±11.85) |
| Women | | | 759 | (63.3) |
| Rural residence | | | 540 | (45.0) |
| Married | | | 521 | (43.4) |
| Household size (mean score) | | | 6.47 | (±5.25) |
| Employed | | | 533 | (44.4) |
| Income (in Ghana Cedis) (mean score) | | | 308.18 | (±338.89) |
| *Level of education* | | | | |
| | Primary/none | | 1034 | (86.2) |
| | Secondary | | 104 | (8.7) |
| | Higher | | 62 | (5.2) |
| Social support (mean score) | | | 6.10 | (±2.68) |
| Alcohol intake | | | 377 | (31.4) |
| Physical activity (mean score) | | | 8.75 | (±4.43) |
| *Self-assessed health* | | | | |
| | Very good | | 239 | (19.9) |
| | Good | | 369 | (30.8) |
| | Fair | | 348 | (29.0) |
| | Poor | | 244 | (20.3) |
| Diagnosed of chronic conditions | | | 636 | (53.0) |
| Functionally impaired (mean score) | | | 13.70 | (±5.09) |

The 10-item Kessler Psychological Distress Scale (K10) was used to quantify non-specific psychological disorders [20]

**Table 2. The associations between food insecurity with hunger and psychological distress score (10–50): Multivariate pooled linear analysis.**

| Variable | Model 1 | | Model 2 | |
|---|---|---|---|---|
| | β | (SE) | β | (SE) |
| *Hunger* (ref: Never/rarely (no food insecurity)) | | | | |
| Sometimes (Moderate food insecurity) | 0.946 | (0.136)*** | 0.705 | (0.160)*** |
| Most of the time/always (Severe food insecurity) | 1.828 | (0.244)*** | 1.813 | (0.280)*** |
| Age (in years) | | | -0.190 | (0.007)** |
| Gender (ref: men) | | | 0.149 | (0.172) |
| Rural/urban residence (ref: rural) | | | -0.206 | (0.148) |
| Marital status (ref: married) | | | 0.637 | (0.159)*** |
| Household size | | | -0.005 | (0.014) |
| Employment status (ref: unemployed) | | | -0.215 | (0.161) |
| Income (in Ghana Cedis) | | | 0.185 | (0.178) |
| Level of education (ref: Primary/none) | | | | |
| Secondary | | | -0.263 | (0.263) |
| Higher | | | -0.045 | (0.315) |
| Social support (ref: high) | | | -0.109 | (0.051)* |
| Alcohol intake (ref: no) | | | 0.080 | (0.167) |
| Physical activity (ref: no) | | | -0.230 | (0.159) |
| *Self-assessed health* (ref: Very good) | | | | |
| Good | | | -0.062 | (0.209) |
| Fair | | | 0.283 | (0.214) |
| Poor | | | 0.914 | (0.243)*** |
| Chronic conditions (ref: no) | | | 0.155 | (0.147) |
| Functionally impaired | | | 0.270 | (0.189) |
| Adjusted $R^2$ | | | 0.212 | |

Coefficients (β) are adjusted for clustering, and robust standard errors are presented in parentheses. The 10-item Kessler Psychological Distress Scale (K10) was used to quantify non-specific psychological disorders [20]. Model 1 was adjusted for age and gender only; Model 2 was adjusted for all potential confounders.

***$p < 0.001$

**$p < 0.005$

*$p < 0.05$.

**Table 3. The associations between food insecurity with hunger and psychological distress score (10–50): Multivariate gender- and age-based linear analysis.**

| | Gender | | | | Age | | | |
|---|---|---|---|---|---|---|---|---|
| | Women | | Men | | 50–64 years | | ≥65 years | |
| Variable | β | (SE) | β | (SE) | β | (SE) | β | (SE) |
| *Hunger* (ref: Never/rarely (no food insecurity)) | | | | | | | | |
| Sometimes (Moderate food insecurity) | 0.736 | (0.198)*** | 0.747 | (0.294)* | 0.848 | (0.221)*** | 0.506 | (0.243)* |
| Most of the time/always (Severe food insecurity) | 1.594 | (0.359)*** | 2.326 | (0.474)*** | 1.478 | (0.368)** | 2.511 | (0.467)*** |
| Adjusted $R^2$ | 0.211 | | 0.249 | | 0.214 | | 0.244 | |

Coefficients (β) are adjusted for clustering, and robust standard errors are presented in parentheses. All sub-group analyses were adjusted for age, gender, rural/urban locality, marital status, household size, employment status, income, level of education, social support, alcohol intake, physical activity, self-rated health, chronic conditions and functional impairment. The 10-item Kessler Psychological Distress Scale (K10) was used to quantify non-specific psychological disorders [20].

***$p < 0.001$

**$p < 0.005$

*$p < 0.05$

evaluated the moderating role of spatial location in the association between food insecurity and psychological distress. However, the interaction analysis did not achieve statistical significance (results are not shown).

## Gender- and age-based models

The gender- and age-wise estimations (Table 3) were performed to investigate any modifying roles of gender and age in the associations of hunger with psychological distress. In Table 3, the odds of psychological distress significantly increased across the levels of hunger for both genders but the effect was higher among men (severe levels of hunger: β = 2.32, SE = 0.474, $p < 0.001$) compare to women (severe levels of hunger: β = 1.59, SE = 0.359, $p < 0.001$). Moreover, psychological distress increased significantly in all age cohorts for those experiencing hunger but the magnitude of severe hunger was higher among ≥65+ age group (β = 2.51, SE = 0.467, $p < 0.001$), compared to those in 50–64 age cohort (β = 1.48, SE = 0.368, $p < 0.005$).

## Discussion and conclusions

This study examined the association between food insecurity with hunger and psychological distress in older age and the possible modifying roles of gender, age and spatial location. The results showed that food insecurity plays a critical role in psychological health and quality of life, broadly, of older people in LMICs. After accounting for the theoretically relevant sociodemographic and health-related factors, food insecurity was positively associated with the risk of psychological distress in this nationally representative sample of low-income older people. The exposure to moderate and severe food insecurity in terms of hunger significantly increased the psychological distress score, providing evidence to support our hypothesis that experiencing hunger will display worse psychological distress in this population.

These results were consistent with a previous study in New Zealand reporting a strong relationship between food insecurity and psychological distress [36]. Despite that research on food insecurity and psychological distress is limited, our findings are similar to previous studies of adults, documenting associations between food insecurity and other major mental disorders including cognitive function [2, 15, 17]. Hadley et al. [37] observed that food insecurity and stressful life events were independently associated with high symptoms of depression, anxiety and post-traumatic stress among Ethiopian adults. Using data from the National Health and Nutrition Examination Survey, Frith & Loprinzi [17] found among older Americans that food insecurity was associated with poor cognitive function. Laraia and colleagues [38] found that food insecurity was not only positively associated with symptoms of depression but also with perceived stress, trait anxiety, and a stronger believe in chance affecting one's life and inversely associated with self-esteem and mastery among expectant women. Among caretakers in rural Tanzania, Hadley & Patil [39] found a strong correlation between food insecurity and anxiety and depression. Leung et al. [40] have also reported a positive relationship of food insecurity with depression. To the best of our knowledge, the findings of the current study appear to be one of the few to present the association between food insecurity *with hunger* and psychological distress among older people in sub-Saharan Africa.

Although specific mechanisms linking food insecurity and psychological distress risk are complex, several hypotheses have been proposed. It has been shown that hunger and the concomitant poor nutritional intake may lead to an increase in stress, strains and body weight which are known modifiable risk factors for various mental instabilities including psychological distress [17, 41]. Further, food insecure older people may experience psychological distress as a result of lack of access to nutritious, affordable, culturally appropriate food and the

inability to feed themselves and their families [36]. It has also been postulated that poor nutrition, stress, and shame in food insecurity may lead to increased risk for mental health problems [9, 12]. As a biological and psychosocial stressor, food insecurity, particularly with hunger, may potentially increase mental and psycho-emotional ill-health [42] particularly during older age.

Our finding that no significant spatial variations exist in the association between food insecurity and psychological distress is notable. This suggests that the severity of food insecurity and its impacts in later life may universally manifest across rural and urban divides [43]. However, consistent with our hypothesis, the association was differentiated by gender and age in the stratified models. Among persons in 65+ age group and men, exposure to severe hunger, especially, was highly related to psychological distress compared to those in the 50–64 age group and women respectively. Thus, this study sheds light on psychological distress, demographics and food insecurity in older-age, an under-explored topic in sub-Saharan Africa with expressive implications for the older population and the policies targeted at healthy aging paradigm. The observation of relative strength of odds suggests that the socioeconomic and health-related covariates explained more of the association between food insecurity and psychological distress in men and older age cohort than women and middle-aged group respectively.

Women generally present higher lifetime mental disorders than men [44]. Interestingly, our observation was contrary to this general assertion and, therefore, appears somewhat difficult to explain. Females tend to have stronger constellation of social networks and support [45] which may reduce the severity of food insecurity, and the associated impacts on psychological distress for women. Furthermore, women tend to report more often than men to feel loved and cared for [27] which can shield them from food insecurity-related mental problems. The age difference may relate well to increased vulnerabilities to brain functioning, a common phenomenon in older people, with the prevalence increasing with age particularly in low-income settings. Growing older reduces hippocampal volumes and this may increase exposure to psychological and cognitive deficits whilst considering the complexities of stressors such as food insecurity [46]. However, this finding is inconsistent with a study in New Zealand in which women had slightly higher odds of the association between food insecurity and psychological distress compared to men [36].

Some limitations of the present study should be recognized when interpreting the findings. Although food insecurity can be measured in a variety of ways, the current analysis was based on hunger and may not directly be comparable to other studies with other measures. Further, all items and measures were based on self-report which could possibly be subject to recall bias. Nevertheless, self-reporting may be the best and certainly most convenient way to capture subjective views. The cross-sectional data used may not include information on how food insecurity evolve over time and how the changes could influence psychological distress in later life. The likelihood of reverse causation deserves attention, given that previous studies have demonstrated that poor health may cause or intensify food insecurity [42]. For more powerful estimations, longitudinal measurements of food insecurity and mental distress indicators are needed to fully understand any potential association of these variables in aging population in Ghana and SSA generally. Despite these limitations, the current study expands previous research by showing how hunger adversely effects psychological distress of aging adults. The key strength lies in the use of a representative sample of unique low-income older people surveyed over a recent time period. Future research would benefit from interactively and longitudinally tracking how food insecurity relates to psychological distress in older age.

In conclusion, the findings emphasize that exposure to hunger and largely, food insecurity may heighten psychological distress risk in later life. Moreover, this association is remarkable in men and older age group. These findings add valuable insights to the scant but growing

literature that points to the need to consider food insecurity interventions in the policy and applied programmatic efforts to reduce psychological distress and its attendant impacts among older people. Crucially, these initiatives should strongly tailor the gender- and age-specific approaches.

## Supporting information

**S1 Data.**
(ZIP)

## Author Contributions

**Conceptualization:** Razak M. Gyasi, Bernard Obeng.

**Data curation:** Razak M. Gyasi, Joseph Y. Yeboah.

**Formal analysis:** Razak M. Gyasi.

**Funding acquisition:** Razak M. Gyasi.

**Investigation:** Razak M. Gyasi.

**Methodology:** Razak M. Gyasi.

**Project administration:** Razak M. Gyasi.

**Resources:** Razak M. Gyasi.

**Software:** Razak M. Gyasi.

**Supervision:** Razak M. Gyasi.

**Validation:** Razak M. Gyasi.

**Visualization:** Razak M. Gyasi.

**Writing – original draft:** Razak M. Gyasi.

**Writing – review & editing:** Razak M. Gyasi, Bernard Obeng, Joseph Y. Yeboah.

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
