## [Decision Letter · Decision Letter 0]

12 Dec 2019

PONE-D-19-29744

Association between Food Insecurity and Psychological Disorders: Results of a Population-based Study in Old Age in Ghana

PLOS ONE

Dear Dr. Gyasi

Thank you for submitting your manuscript to PLOS ONE. After careful consideration, we feel that it has merit but does not fully meet PLOS ONE’s publication criteria as it currently stands. Therefore, we invite you to submit a revised version of the manuscript that addresses the points raised during the review process.

You should addres all the suggestions made by the reviewers, but in particular you should address the conceptual and methodological issues as well as to improve the quality of your discussion section.

We would appreciate receiving your revised manuscript by January 15, 2020. To enhance the reproducibility of your results, we recommend that if applicable you deposit your laboratory protocols in protocols.io, where a protocol can be assigned its own identifier (DOI) such that it can be cited independently in the future. For instructions see: http://journals.plos.org/plosone/s/submission-guidelines#loc-laboratory-protocols

We look forward to receiving your revised manuscript.

Kind regards,

Berta Schnettler

Academic Editor

PLOS ONE

Journal Requirements:

1. We note that you have indicated that data from this study are available upon request. PLOS only allows data to be available upon request if there are legal or ethical restrictions on sharing data publicly. For information on unacceptable data access restrictions, please see http://journals.plos.org/plosone/s/data-availability#loc-unacceptable-data-access-restrictions.

Reviewers' comments:

Reviewer's Responses to Questions

**Comments to the Author**

1. Is the manuscript technically sound, and do the data support the conclusions?

Reviewer #1: Partly

Reviewer #2: Partly

2. Has the statistical analysis been performed appropriately and rigorously? 

Reviewer #1: Yes

Reviewer #2: Yes

3. Have the authors made all data underlying the findings in their manuscript fully available?

Reviewer #1: Yes

Reviewer #2: Yes

4. Is the manuscript presented in an intelligible fashion and written in standard English?

Reviewer #1: Yes

Reviewer #2: No

5. Review Comments to the Author

Reviewer #1: Referee Report for “Association between Food Insecurity and Psychological Disorders...”

To the best of my knowledge, this topic hasn’t been addressed in an older population in a low-income country (LIC). As such, it can make a contribution. But, a few things which will dramatically shorten the length of the paper:

1. The paper takes far too long to get to its central point, namely that food insecurity can lead to psychological disorders among seniors in Ghana. The introduction should be about three paragraphs – why this is an issue, how this paper contributes to our understanding, and the central findings.

2. There is only one convincing measure of food insecurity, the first one the authors mention, the one they call “hunger”. The other two measures should be dropped insofar as they may measure very different things and, in particular, different things than what is usually meant by food insecurity. As an example, lots of people with sufficient financial resources do not eat breakfast.

3. The Discussion should concentrate on how this paper contributes to our understanding of the impact of food insecurity on psychological disorders in an LIC, in particular, for seniors. As is stands now, there is too much extraneous information provided.

Reviewer #2: I would argue that hunger and food insecurity are not “eating disorders”, and yet, the very first sentence of the paper speaks of eating disorders and its implications for mental health. This sets the stage for the reader, in a very different direction than the one the paper is aimed at. Eating disorders occur in the presence of abundant or sufficient food, not in its absence.

Even though the English Grammar and Syntax are, for the most part correct, the phrasing of the text tends to become cumbersome, at times, and some word-choices and verb forms, could gain precision. The manuscript would benefit from the review by a specialized editor, before resubmitting it.

It is not clear to this reader whether the data were collected freshly, and specifically, for the study being reported here, or whether they are secondary data collected in the context of a prior, broader, and longitudinal research program. It would perhaps be necessary to clarify this condition more explicitly.

The displays of results by histograms in Figures 1, 2, and 3, are either wrong, or they need a better description, in order to be understood correctly. The gray bars, at the right-hand side in every set of bars are labeled “Total”; however, if anything, they seem to represent an ‘average’ between the prevalence of “distressed” and “not distressed” participants, within a specific “food insecurity” condition. But, if this were the case, averaging percentages is not correct.

Another pair of equivocally equated terms in this manuscript are “psychological disorders” (PD) and “psychological distress”. The first category encompasses a wide range of psychopathological and psychiatric disorders; whereas, the second, involves only anxiety and depressive disorders, which are the kinds of disorders that were actually assessed. In spite of this, every time the initials PD are used, during the discussion, it refers to “psychological disorders”, not distress.

The lack of accuracy in the use of these two terms, lends itself to important distortions in the interpretation of results. Such distortions may provide grounds for a policy statement such as the one it was suggested, at the end of the paper, that providing food to the needy elderly might help solve the mental health problems of this population. Of course, if one provided food to the hungry, and gave certainty that food will continue to be available to them, their anxiety levels would decrease considerably; but, those who suffered from zchizophrenia or dementia whould hardly be alleviated. So, properly, it is not a matter of mental health.

6. PLOS authors have the option to publish the peer review history of their article (what does this mean?). If published, this will include your full peer review and any attached files.

Reviewer #1: No

Reviewer #2: No

---

## [Author Response · Author response to Decision Letter 0]

8 Jan 2020

RESPONSES TO COMMENTS

Reviewer #1.

Referee Report for “Association between Food Insecurity and Psychological Disorders...”

To the best of my knowledge, this topic hasn’t been addressed in an older population in a low-income country (LIC). As such, it can make a contribution. But, a few things which will dramatically shorten the length of the paper:

Response: We would like to thank the reviewer for these critical observations and positive comments on our manuscript. 

1. The paper takes far too long to get to its central point, namely that food insecurity can lead to psychological disorders among seniors in Ghana. The introduction should be about three paragraphs – why this is an issue, how this paper contributes to our understanding, and the central findings.

Response: The introduction has been re-written and shortened taking into consideration the concerns of the reviewer.

2. There is only one convincing measure of food insecurity, the first one the authors mention, the one they call “hunger”. The other two measures should be dropped insofar as they may measure very different things and, in particular, different things than what is usually meant by food insecurity. As an example, lots of people with sufficient financial resources do not eat breakfast.

Response: Hunger has been captured as the only measure of food insecurity in the revised version of the paper. The other measures included initially, namely, breakfast skipping and timing of the first daily meal have been dropped. 

3. The Discussion should concentrate on how this paper contributes to our understanding of the impact of food insecurity on psychological disorders in an LIC, in particular, for seniors. As it stands now, there is too much extraneous information provided.

Response: The discussion of the paper has been re-written to concentrate the explanation and implications of the main study findings. The strenuous issues and information have been removed. 

Reviewer #2.

I would argue that hunger and food insecurity are not “eating disorders”, and yet, the very first sentence of the paper speaks of eating disorders and its implications for mental health. This sets the stage for the reader, in a very different direction than the one the paper is aimed at. Eating disorders occur in the presence of abundant or sufficient food, not in its absence.

Response: Thanks to the reviewer for this important observation and comment. In the revised manuscript, the entire introduction has been re-considered and therefore has taken care of these changes. We removed “eating disorders” from the manuscript altogether.

Even though the English Grammar and Syntax are, for the most part correct, the phrasing of the text tends to become cumbersome, at times, and some word-choices and verb forms, could gain precision. The manuscript would benefit from the review by a specialized editor, before resubmitting it.

Response: We have checked sections of the manuscript to rectify grammatical and typographical errors. Thorough proof reading and editing have been done throughout to correct these errors.

It is not clear to this reader whether the data were collected freshly, and specifically, for the study being reported here, or whether they are secondary data collected in the context of a prior, broader, and longitudinal research program. It would perhaps be necessary to clarify this condition more explicitly.

Response: This study forms part of a larger, original study that aimed at generating comprehensive data on socioeconomic, health and wellbeing profile of older people in Ghana and provided a basis for comprehensive analysis of many facets of aging in the LMICs. The variables in the current study were drawn from this larger study. This clarification has been provided in the revised version of the paper.

The displays of results by histograms in Figures 1, 2, and 3, are either wrong, or they need a better description, in order to be understood correctly. The gray bars, at the right-hand side in every set of bars are labeled “Total”; however, if anything, they seem to represent an ‘average’ between the prevalence of “distressed” and “not distressed” participants, within a specific “food insecurity” condition. But, if this were the case, averaging percentages is not correct.

Response: Figures 2 and 3 have been removed from the manuscript based on the first reviewer’s suggestion to drop breakfast skipping and timing of first daily meal as measures of food insecurity from the manuscript. While Figure 1 was checked to be correct, we have chosen to drop it too. 

Another pair of equivocally equated terms in this manuscript are “psychological disorders” (PD) and “psychological distress”. The first category encompasses a wide range of psychopathological and psychiatric disorders; whereas, the second, involves only anxiety and depressive disorders, which are the kinds of disorders that were actually assessed. In spite of this, every time the initials PD are used, during the discussion, it refers to “psychological disorders”, not distress.

Response: We agree with the reviewer for the comment. The use of the term “psychological disorders” and “mental disorders” have all been removed in the revised manuscript. Instead, the term “psychological distress” has been used throughout the draft.

The lack of accuracy in the use of these two terms, lends itself to important distortions in the interpretation of results. Such distortions may provide grounds for a policy statement such as the one it was suggested, at the end of the paper, that providing food to the needy elderly might help solve the mental health problems of this population. Of course, if one provided food to the hungry, and gave certainty that food will continue to be available to them, their anxiety levels would decrease considerably; but, those who suffered from zchizophrenia or dementia would hardly be alleviated. So, properly, it is not a matter of mental health. 

Response: We thank the reviewer for these observations and we appreciate this critical comment. We have used the “psychological distress” in all aspect of the manuscript. 

Again, we would like to thank the Editor and the two anonymous reviewers for the opportunity to revise and resubmit our paper to PloS One. 

Yours sincerely,

Corresponding author

---

## [Decision Letter · Decision Letter 1]

12 Feb 2020

PONE-D-19-29744R1

Impact of Food Insecurity with Hunger on Mental Distress among Community-Dwelling Older Adults

PLOS ONE

Dear Mr. Gyasi,

Thank you for submitting your manuscript to PLOS ONE. After careful consideration, we feel that it has merit but does not fully meet PLOS ONE’s publication criteria as it currently stands. Therefore, we invite you to submit a revised version of the manuscript that addresses the points raised during the review process.

Please address the language requirements made by the reviewer.

We would appreciate receiving your revised manuscript by Mar 27 2020 11:59PM. To enhance the reproducibility of your results, we recommend that if applicable you deposit your laboratory protocols in protocols.io, where a protocol can be assigned its own identifier (DOI) such that it can be cited independently in the future. For instructions see: http://journals.plos.org/plosone/s/submission-guidelines#loc-laboratory-protocols

We look forward to receiving your revised manuscript.

Kind regards,

Berta Schnettler

Academic Editor

PLOS ONE

Reviewers' comments:

Reviewer's Responses to Questions

**Comments to the Author**

1. If the authors have adequately addressed your comments raised in a previous round of review and you feel that this manuscript is now acceptable for publication, you may indicate that here to bypass the “Comments to the Author” section, enter your conflict of interest statement in the “Confidential to Editor” section, and submit your "Accept" recommendation.

Reviewer #1: All comments have been addressed

Reviewer #2: All comments have been addressed

2. Is the manuscript technically sound, and do the data support the conclusions?

Reviewer #1: Yes

Reviewer #2: Yes

3. Has the statistical analysis been performed appropriately and rigorously? 

Reviewer #1: Yes

Reviewer #2: Yes

4. Have the authors made all data underlying the findings in their manuscript fully available?

Reviewer #1: Yes

Reviewer #2: Yes

5. Is the manuscript presented in an intelligible fashion and written in standard English?

Reviewer #1: Yes

Reviewer #2: Yes

6. Review Comments to the Author

Reviewer #1: (No Response)

Reviewer #2: The revised version of the manuscript appropriately incorporates all my comments to the initial version. However, the new text still carries a some mishandling of the English language and a few verb-form errors.

For example, on page 5, last paragraph, second sentence, it is nor clear what is meant by "...self-reported diagnosis by a health profesional... " I presume that the participants' health condition can either be self-reported, or diagnosed by a health professional. AS I understood the sentence, health profesionals' self-reported diagnosis, necessarily refers to their own health condition, not to that of partricipants...

Passages like this one should be found and corrected

7. PLOS authors have the option to publish the peer review history of their article (what does this mean?). If published, this will include your full peer review and any attached files.

Reviewer #1: No

Reviewer #2: No

---

## [Author Response · Author response to Decision Letter 1]

13 Feb 2020

RESPONSES TO COMMENTS

Reviewer #2.

The revised version of the manuscript appropriately incorporates all my comments to the initial version. However, the new text still carries some mishandling of the English language and a few verb-form errors.

For example, on page 5, last paragraph, second sentence, it is not clear what is meant by "...self-reported diagnosis by a health professional... ". I presume that the participants' health condition can either be self-reported, or diagnosed by a health professional. As I understood the sentence, health professionals' self-reported diagnosis, necessarily refers to their own health condition, not to that of participants.

Passages like this one should be found and corrected 

Response: Very many thanks to Reviewer #2 for this important observations and the comment on the English usage in the manuscript. The entire draft has been checked and edited thoroughly for grammar. Extensive proof reading and editing have been done to rectify grammatical errors.

I thank Editor and the reviewer for the opportunity to revise and resubmit the paper for a consideration by PloS One. 

Yours sincerely,

Razak Gyasi

---

## [Editor Report · Decision Letter 2]

18 Feb 2020

Impact of Food Insecurity with Hunger on Mental Distress among Community-Dwelling Older Adults

PONE-D-19-29744R2

Dear Razak Mohammed Gyasi

We are pleased to inform you that your manuscript has been judged scientifically suitable for publication and will be formally accepted for publication once it complies with all outstanding technical requirements.

With kind regards,

Berta Schnettler

Academic Editor

PLOS ONE
---

## [Editor Report · Acceptance letter]

16 Mar 2020

PONE-D-19-29744R2 

Impact of Food Insecurity with Hunger on Mental Distress among Community-Dwelling Older Adults 

Dear Dr. Gyasi:

I am pleased to inform you that your manuscript has been deemed suitable for publication in PLOS ONE. Congratulations! Your manuscript is now with our production department. 

With kind regards,

on behalf of

Dr. Berta Schnettler 

Academic Editor

PLOS ONE